# Systematic Analysis of Driving Modes and NiFe Layer Thickness in Planar Hall Magnetoresistance Sensors

**DOI:** 10.3390/s25041235

**Published:** 2025-02-18

**Authors:** Changyeop Jeon, Mijin Kim, Jinwoo Kim, Sunghee Yang, Eunseo Choi, Byeonghwa Lim

**Affiliations:** 1Department of Physics and Chemistry, Daegu Gyeongbuk Institute of Science and Technology (DGIST), Daegu 42988, Republic of Korea; jco4268@dgist.ac.kr (C.J.); kmj@dgist.ac.kr (M.K.); k306kjw@dgist.ac.kr (J.K.); didtmdgml1105@jejunu.ac.kr (S.Y.); cesok@khu.ac.kr (E.C.); 2Department of Smart Sensor Engineering, Andong National University, Andong-si 36729, Republic of Korea

**Keywords:** magnetoresistive sensors, driving mode, constant current mode, constant voltage mode

## Abstract

Planar Hall magnetoresistance (PHMR) sensors are widely utilized due to their high sensitivity, simple structure, and cost-effectiveness. However, their performance is influenced by both the driving mode and the thickness of the ferromagnetic layer, yet the combined effects of these factors remain insufficiently explored. This study systematically investigates the impact of Ni_80_Fe_20_ thickness (5–35 nm) on PHMR sensor performance under constant current (CC) and constant voltage (CV) modes, with a focus on optimizing the peak-to-peak voltage (V_p-p_). In CC mode, electron surface scattering at 5–10 nm increases resistance, leading to a sharp rise in V_p-p_, followed by a decline as the thickness increases. In contrast, CV mode minimizes resistance-related effects, with sensor signals predominantly governed by magnetization-dependent resistivity. Experimentally, the optimal V_p-p_ was observed at 25 nm in CV mode. However, for thicknesses beyond this point, the reduction in sensor resistance suggests that voltage distribution across both the sensor and external load resistance significantly influences performance. These findings provide practical insights into optimizing PHMR sensors by elucidating the interplay between driving modes and material properties. The results contribute to the advancement of high-performance PHMR sensors with enhanced signal stability and sensitivity for industrial and scientific applications.

## 1. Introduction

Magnetic sensors have become increasingly integrated into various electronic devices, playing a critical role across various fields [1,2,3]. These sensors are not only used in electronic compasses [4] and navigation systems, which rely on detecting the Earth’s magnetic field, but also serve as motion recognition sensors in virtual reality (VR) environments [5]. Additionally, they are essential components in advanced technologies such as biosensors [6,7,8], tactile sensors for robotics [9], magnetic memory devices [10], and spintronic devices [11]. Magnetic sensors are typically categorized based on their operating principles into semiconductor-based Hall [12,13] and magnetoresistive sensors [14].

Magnetoresistive sensors are classified into several types based on their physical principles: anisotropic magnetoresistance (AMR) [15], giant magnetoresistance (GMR) [16], tunneling magnetoresistance (TMR) [17], and planar Hall magnetoresistance (PHMR) [18]. GMR- and TMR-based sensors are commonly used in high-sensitivity applications due to their high magnetoresistance ratio, but relatively high noise levels can limit their performance. In contrast, PHMR sensors offer the advantages of a simple structure, the ease of fabrication, and their cost-effectiveness [19]. In particular, PHMR sensors provide excellent linear response to magnetic fields, high signal sensitivity, low offset voltage, and a superior signal-to-noise ratio [20]. Additionally, PHMR sensors exhibit remarkable stability against ambient temperature changes, which has led to their increasing use in various applications.

The performance of PHMR sensors is determined by the shape and structure of the sensor’s thin films. Various designs, such as cross-shaped [21], tilted [22], and Wheatstone bridge-type sensors [23], have been studied to maximize the sensor’s output voltage. Structurally, magnetic sensors that respond linearly to magnetic fields free from hysteresis have been developed through a bi-layer structure based on exchange bias between ferromagnetic and antiferromagnetic materials [24]. Furthermore, research has been conducted on optimizing antiferromagnetic materials to enhance sensor sensitivity or improve thermal stability by incorporating a non-magnetic layer between the ferromagnetic and antiferromagnetic layers [25,26,27].

In sensors using a single-layer NiFe structure, the output signal is determined by the current density and magnetoresistance properties, regardless of the driving mode (constant current (CC) or constant voltage (CV)). However, for multi-layer magnetoresistive sensors, the driving mode influences both the current density and input voltage in the NiFe layer, leading to differences in sensor behavior. While significant research has focused on optimizing the ferromagnetic materials that determine sensor performance, most studies have primarily investigated the CC mode, particularly with respect to variations in the thickness of the NiFe layer [28]. On the other hand, structural optimization studies for the CV mode, which is widely used in industrial applications, remain insufficiently explored.

This study systematically analyzed the performance and characteristics of bi-layer PHMR sensors across different driving modes by varying the thickness of the NiFe layer between 5 nm and 35 nm in both CC and CV modes. Sensor operational characteristics were evaluated, and current density simulations were conducted to understand how NiFe thickness impacts sensor performance and current density distribution. The magnetic properties as a function of NiFe thickness were also analyzed to assess their influence on the sensor’s operating range and sensitivity, offering design guidelines for optimal performance. This research presents strategies for enhancing PHMR sensor performance and contributes to high-sensitivity magnetic sensor development by optimizing driving modes across thickness conditions. The findings will provide foundational data for future spintronic device design and application.

## 2. Materials and Methods

A cross-patterned bi-layer PHMR sensor, shown in Figure 1a, was fabricated to analyze its characteristics according to the driving mode. The SiO_2_ substrate was produced through a wet oxidation process, and the thin films were deposited using sputtering in the sequence of Ta (5 nm)/Ni_80_Fe_20_ (x)/Ir_25_Mn_75_ (10 nm)/Ta (5 nm). Here, Ta served as both the seed and capping layers. Ni_80_Fe_20_ was selected as the active ferromagnetic layer due to its excellent soft magnetic properties, including high magnetic permeability for rapid response to external magnetic fields and low coercivity to minimize magnetic hysteresis. Ir_25_Mn_75_ was chosen as the antiferromagnetic layer with a thickness of 10 nm to induce exchange bias and align the magnetization of the Ni_80_Fe_20_ layer along the easy axis. This configuration minimizes hysteresis along the hard axis, ensuring a linear magnetic response and improving the sensor’s reliability and performance [28]. The thickness of the Ni_80_Fe_20_ layer was varied from 5 nm to 35 nm, reaching a saturation point for Δ*ρ*. The sensor’s cross shape was designed with line and width dimensions set to 200 μm each, and current application electrodes were placed along the x-axis with voltage measurement electrodes along the y-axis for sensor operation and signal measurement.

The bi-layer thin film structure was deposited using a DC magnetron sputtering method, and a constant external magnetic field of 250 Oe was applied during deposition to induce the exchange bias direction of the Ni_80_Fe_20_ and Ir_25_Mn_75_ layers. The base pressure was carried out under an Ar (99.999%) atmosphere at a pressure of 3 × 10^−3^ Torr. The applied voltage during the sputtering process ranged from 200 V to 250 V, depending on the target material. Specifically, the sputtering power was set to 50 W to ensure optimized deposition rates and uniformity. UV lithography and lift-off processes formed the sensor’s cross pattern and the electrode patterns. Figure 1b shows the structure of the fabricated cross-patterned PHMR sensor.

For the characterization of the fabricated sensor, measurements were performed under different driving modes using a 2400 Keithley source meter (Keithley Instruments, Cleveland, OH, USA) and an HP 34401A multimeter (HP, Palo Alto, CA, USA) controlled by a LabVIEW program. In the constant current mode, a DC of 1 mA was applied to the sensor, while in the constant voltage mode, a voltage of 1 V was applied. A magnetic field was applied using Helmholtz coils, ranging from +300 Oe to −300 Oe, to analyze the magnetic characteristics relative to the field direction. The noise evaluation of the cross-type PHMR sensor was conducted using Agilent’s 35670A Spectrum Analyzer (Agilent Technologies, Santa Clara, CA, USA). A vibrating sample magnetometer (VSM) system (Lake Shore Cryotronics 7407, Lake Shore Cryotronics Inc., Westerville, OH, USA) was used to measure the hysteresis curves along the easy and hard magnetization axes.

ANSYS Maxwell Electronics software (2021 R1) was used to simulate the current density changes with varying Ni_80_Fe_20_ thickness, analyzing the current density distribution for the same bi-layer structure as the fabricated sensor. The simulation compared current density changes between the CC and CV modes, providing insights into how Ni_80_Fe_20_ thickness affects current density characteristics and allows for an analysis of the differences in sensor characteristics according to the driving mode.

## 3. Background for the Relationship of PHMR Sensor

PHMR and AMR refer to phenomena in which the resistance changes depending on the angle between the current and the magnetization direction in ferromagnetic materials, a behavior attributed to the spin–orbit interaction. When the magnetic field is applied, the orbital electrons are distorted, leading to energy exchange between the spin and orbital moments, which alters the electron energy band structure and electrical properties. These changes affect the direction of electron movement, causing the resistance to vary depending on the angle between the magnetic field and the current direction [19]. Therefore, the signal characteristics of AMR and PHMR can be explained based on Ohm’s law as follows:(1)$$\vec{E}=\rho \left(\hat{\phi }\right)=\left[\begin{array}{cc} \rho_{xx} & \rho_{xy}\\ \rho_{yx} & \rho_{yy} \end{array}\right]\vec{J}\;with\;\left\{\begin{array}{c} \rho_{xx}\left(\phi \right)=\rho_{⊥}+\Delta \rho \cos^{2}\phi \\ \rho_{yx}\left(\phi \right)=\rho_{xy}\left(\phi \right)=\frac{\Delta \rho }{2}\sin2\phi \\ \rho_{yy}\left(\phi \right)=\rho_{⊥}+\Delta \rho \sin^{2}\phi \end{array}\right.(H_{s})$$

Here, *ρ*_∥_ and *ρ*_⊥_ represent the resistivity parallel and perpendicular to the magnetization direction, respectively, with Δ*ρ* denoting the difference between them. This resistivity difference contributes to a component of the electric field in the direction of the current. The sensor signal produced by this mechanism depends on the relative angle *ϕ* between the magnetization and current directions.(2)$$\left\{\begin{array}{c} V_{x}=lJ_{x}(\rho_{⊥}+\Delta \rho \cos^{2}\phi )\\ V_{y}=wJ_{x}\frac{\Delta \rho }{2}\sin2\phi \end{array}\right.$$

Here, *V_x_* represents the AMR signal caused by the diagonal resistance component *ρ_xx_*, while *V_y_* denotes the PHMR signal generated by the off-diagonal resistance component *ρ_xy_*.

Both AMR and PHMR sensors share a common mechanism where their output signals are determined by the *J* and Δ*ρ*. However, there are notable differences between the two. AMR sensors typically exhibit higher output signals compared to PHMR sensors, but their response is nonlinear at zero magnetic field due to intrinsic offset voltages. In contrast, the PHMR signal is characterized by its linear response to the applied magnetic field and the absence of offset voltages. This distinction makes PHMR sensors particularly advantageous in applications requiring high precision and linearity in detecting magnetic field variations.

PHMR sensors operate within a range of *ϕ* from −45° to +45°, depending on the magnetic field intensity. The sensor achieves its maximum output voltage within this range, as Figure 1c depicts. The maximum output voltage within the sensor’s operating range (H_p-p_) is given by the following equation:(3)$$V_{\mathrm{p-p}}=wJ_{x}\Delta \rho$$

Therefore, regardless of the driving mode of the PHMR sensor, its performance is determined by *J_x_* and Δ*ρ* of the NiFe layer. Additionally, the sensor’s sensitivity is defined by the change in the sensor signal in response to various in the magnetic field.

Δ*ρ* is one of the key factors determining a PHMR sensor’s output performance. The larger the Δ*ρ*, the higher the signal difference and sensitivity the magnetoresistive sensor can achieve for the same magnetic field magnitude. The magnitude of Δ*ρ* depends on the thickness of the FM layer and can be quantitatively described according to the Funch-Sondheimer theory as follows [29]:(4)$$\rho =\rho_{0}(1+\frac{3}{8\kappa }),\;\kappa =\frac{t}{\lambda }$$
where *ρ* and *ρ*_0_ represent the resistivity of the metal in its thin film and bulk state, respectively, *λ* denotes the mean free path of the thin film, and t represents its thickness. The mean free path and bulk resistivity are influenced by the direction of the applied current and the magnetization direction of the Ni_80_Fe_20_ layer.

Appendix A presents the analysis of resistivity changes with varying FM layer thickness. The resistivity tends to decrease inversely with increasing thickness, mainly exhibiting a sharp increase as the thickness decreases. This phenomenon is attributed to the pronounced electron scattering effects in thin films. As the film thickness decreases, the mean free path of electrons is reduced, leading to a significant increase in resistivity. Appendix A visualizes the relative change in Δ*ρ*, which tends to saturate as the thickness increases. The saturation of Δ*ρ* indicates that further improvements in resistivity become limited beyond a certain thickness. This underscores the importance of selecting an optimal FM layer thickness to enhance the sensor’s performance.

## 4. Results

### 4.1. Analysis of Current Density

The output performance of the PHMR sensor, as shown in Equation (3), is determined by the current density flowing through the Ni_80_Fe_20_ layer and the resistivity change due to variations in the magnetic field. The sensor operates primarily in CC mode or CV mode, where the current density distribution varies depending on the operating mode and the thickness of the active sensor layer. In particular, for multi-layer thin-film sensors, the resistivity and thickness of each layer significantly influence the overall resistance, resulting in variations in current distribution across the layers depending on the driving mode (CC or CV). These differences in current density suggest that the optimal thin-film structure for enhancing output performance may vary depending on the operating mode.

A simulation was performed, using ANSYS Maxwell to examine the current density distribution as a function of Ni_80_Fe_20_ layer thickness to quantitatively analyze the changes in current density according to the operating mode. The simulation calculated the current density distribution based on the resistivity values of Ta, Ni_80_Fe_20_, and Ir_25_Mn_75_, which constitute the bi-layer structure [30]. In CC mode, a current of 1 mA was applied through the electrodes, while in CV mode a voltage of 1 V was applied. The changes in current density distribution for each mode were analyzed, providing insights into the need for an optimized thin-film structure design tailored to each operating mode.

In CC mode, the variation in the thickness of the NiFe layer affects the current density, which the following equation can express:(5)$$J_{NiFe(CC)}=\frac{\rho_{buffer}}{w(t_{buffer}\rho_{FM}+t_{FM}\rho_{buffer})}I_{sensor}$$

Here, *I_sensor_* represents the current applied to the sensor, *ρ_FM_* and *t_FM_* denote the resistivity and thickness of the Ni_80_Fe_20_ layer, respectively, *ρ_buffer_* is the resistivity of the buffer layer, and *w* represents the width of the sensor.

Figure 2a shows the simulation results of the current density distribution in CC mode as a function of the Ni_80_Fe_20_ layer thickness. The simulation results indicate that the density distribution changes as the thin-film thickness increases. Notably, at a thickness of 5 nm, a relatively higher current density was observed, and as the thickness increased, the current density gradually decreased. This pattern is similarly observed in the buffer layers, such as Ta and Ir_25_Mn_75_, as shown in Figure 2c.

When the sensor operates in CV mode, the buffer and Ni_80_Fe_20_ layers theoretically receive the same applied voltage. In this mode, the input voltage remains constant, and the current density in the Ni_80_Fe_20_ layer can be expressed as follows:(6)$$J_{NiFe(CV)}=\frac{V_{sensor}}{l\rho_{FM}}$$
where the *V_sensor_* represents the voltage applied to the sensor, and the sensor’s output performance in CV mode is influenced more by the magnitude of the driving voltage and the resistivity rather than the thickness of the Ni_80_Fe_20_ layer. The applied voltage magnitude and the material properties, such as Δ*ρ* of the Ni_80_Fe_20_ layer, become more critical factors for determining the sensor’s sensitivity

Figure 2b presents the simulation results of current density distribution in CV mode as a function of Ni_80_Fe_20_ layer thickness. The analysis shows that higher current density regions were observed when the thin films were all relatively thin, indicating that in CV mode the current density in the Ni_80_Fe_20_ layer is less sensitive to thickness variations compared to CC mode. Furthermore, as shown in Figure 2d, this behavior is consistent across other buffer layers, demonstrating minimal influence from thickness variations. Therefore, in CC mode the current density is significantly affected by changes in thin-film thickness due to the constant current. However, in CV mode the applied voltage determines the current density more strongly than the thin film’s thickness, as the voltage is distributed proportionally across the layers depending on their resistances.

### 4.2. Characteristics of PHMR Sensor in CC Mode

CC mode is one of the driving methods for sensors, maintaining a constant current regardless of external resistance or temperature changes connected to the device. In the CC mode of the PHMR sensor, the output signal is determined by the interaction between the current density and Δ*ρ* as the thickness of the Ni_80_Fe_20_ layer changes. As shown in Figure 2a, when the Ni_80_Fe_20_ layer becomes thicker, the current density decreases while Δ*ρ* increases, contributing to signal amplification. These two factors are crucial in determining the sensor’s final output characteristics. The output signal of a PHMR sensor operating in CC mode can be expressed as follows:(7)$$V_{PHMR(CC)}=\frac{\rho_{buffer}I_{sensor}}{t_{buffer}\rho_{FM}+t_{FM}\rho_{buffer}}\frac{\Delta \rho }{2}\sin2\phi$$

Figure 3a illustrates the variation in the output signal of the PHMR sensor as a function of Ni_80_Fe_20_ layer thickness in the CC mode. The analysis reveals that the sensor’s output signal exhibits an asymmetric pattern depending on the Ni_80_Fe_20_ layer thickness. As the thickness increases, the V_p-p_ initially increases sharply from 5 nm to 10 nm, but tends to saturate beyond a certain thickness. This rapid increase in V_p-p_ for very thin Ni_80_Fe_20_ layers can be explained by the Fuchs–Sondheimer theory [29]. In extremely thin layers (e.g., 5 nm), electron surface scattering has a significant impact on resistivity, resulting in a lower Δ*ρ*. As the layer thickness increases to 10 nm, the effect of surface scattering is reduced, leading to more stable resistivity and an enhanced Δ*ρ*. This contributes to the observed sharp rise in V_p-p_. Beyond this thickness, however, the losses in current density begin to counteract the increasing Δ*ρ*, leading to saturation in the output signal.

As the thickness of the Ni_80_Fe_20_ layer increases, the sensor’s output signal characteristics change in both the output voltage magnitude and the shape of the characteristic curve. Figure 3a illustrates that, although the output voltage decreases as the Ni_80_Fe_20_ layer thickens, H_p-p_ also decreases, which improves the sensor’s sensitivity. This decrease in H_p-p_ with increased Ni_80_Fe_20_ thickness suggests that the sensor becomes more responsive to subtle changes in the magnetic field.

The variation in Ni_80_Fe_20_ layer thickness affects the bi-layer structure’s magnetic properties and the PHMR sensor’s operating characteristics. Figure 4a,b represents the magnetic hysteresis curve along the easy and hard axis, respectively, showing that the exchange bias weakens as the Ni_80_Fe_20_ thickness gradually increases, and domain rotation tends to occur more abruptly within a narrower magnetic field range. This magnetization change occurs due to the interaction between the ferromagnetic and antiferromagnetic layers at the interface.

Additionally, Figure 4c illustrates the decrease in exchange bias (H_ex_) with increasing Ni_80_Fe_20_ thickness, which is a pivotal aspect of the observed magnetic behavior. This phenomenon indicates that H_ex_ is inversely proportional to the thickness of the ferromagnetic (FM) layer. When the Ni_80_Fe_20_ layer is thin, the coupling between the FM and antiferromagnetic (AFM) layers is strong, leading to a pronounced exchange bias. However, as the thickness increases, the bulk properties of the FM layer take precedence, reducing the influence of interfacial spins and weakening the exchange anisotropy, thus decreasing H_ex_.

In particular, as the thickness increases, the exchange bias weakens during the magnetization process, reducing the sensor’s H_p-p_, as illustrated in Figure 4d. This results in enhanced sensitivity. These findings explain the influence of Ni_80_Fe_20_ layer thickness on the sensor’s operating characteristics and sensitivity, providing valuable guidelines for optimizing the thickness design for improved sensor performance.

### 4.3. Characteristics of PHMR Sensor in CV Mode

In CV mode, a uniform voltage is applied across the entire sensor, allowing the current density to remain consistent regardless of the thickness variation in the thin layers. The output signal of a PHMR sensor operating in CV mode can be expressed as follows:(8)$$V_{PMMR(CV)}=w\frac{V_{sensor}}{l\rho_{FM}}\frac{\Delta \rho }{2}\sin2\phi$$

In contrast to CC mode, the output signal in CV mode is influenced by the geometric properties, such as width, and the changes in Δ*ρ* due to the thickness of the NiFe layer. As a result, according to Equation (8), the PHMR sensor signal is expected to increase with the NiFe layer thickness in CV mode, as depicted in Appendix A.

Figure 5a presents the PHMR characteristics of the bi-layer structure in CV mode for various Ni_80_Fe_20_ thicknesses. As the Ni_80_Fe_20_ layer thickness increases, the initial output performance of the sensor improves due to the increase in Δ*ρ*, as there is no current density loss. However, as shown in Figure 5b, the sensor’s output signal decreases when the Ni_80_Fe_20_ layer exceeds 25 nm in thickness due to the Δ*ρ* approaching its saturation point. As indicated in Equation (8), as the thickness of the NiFe layer increases, the overall resistance of the sensor decreases, leading to a reduction in the applied voltage across the sensor, excluding the electrodes. This effect becomes more pronounced at greater thicknesses, ultimately contributing to the decline in the sensor’s output signal.

To further characterize the sensor performance, we evaluated the equivalent magnetic noise and limit of detection (LOD) for both CC and CV modes. For CC mode, the optimal Ni_80_Fe_20_ thickness is 10 nm, achieving a sensitivity of 0.859 µV/Oe, as shown in Table 1, while for CV mode, the optimal thickness is 25 nm, with a sensitivity of 13.06 µV/Oe, as shown in Table 2. Additionally, the noise spectrum density of a cross-type PHMR sensor, presented in Appendix A, indicates a noise level of approximately 2.4 nV/√Hz, resulting in a magnetic resolution of 279 nT/√Hz for the 10 nm thickness and 18.4 nT/√Hz for the 25 nm thickness.

The findings of this study suggest that in CV mode, the effect of Ni_80_Fe_20_ layer thickness on the sensor output signal is influenced not only by Δ*ρ* but also by the resistance ratio between the sensor and the electrodes. This effect occurs because, as the Ni_80_Fe_20_ layer thickness increases, the sensor’s resistance decreases, becoming relatively lower than that of the sensor electrodes. Consequently, the input voltage applied to the sensor decreases due to voltage distribution within the sensor circuit. The following equation can express this voltage distribution:(9)$$V_{sensor}=V_{source}\frac{R_{Load}}{R_{sensor}+R_{Load}}$$
*V_sensor_* is the voltage applied to the sensor, *R_sensor_* is the resistance of the Ni_80_Fe_20_ layer, *R_Load_* is the resistance of the electrode, and *V_source_* is the total input voltage applied to the system.

To evaluate the effect of load resistance based on the sensor driving mode, an external resistance in the 50 to 500 Ω was connected to the PHMR sensor, and its characteristics were analyzed. Appendix A shows the characteristics curves of the PHMR sensor considering the external resistance and variations in Ni_80_Fe_20_ layer thickness. In CC mode, the same current is applied to the sensor regardless of the load resistance, which means that, as shown in Figure 6a, the sensor’s performance is influenced by the reduction in current density depending on the thickness and type of thin films that compose the sensor. As a result, the sensor’s output performance is primarily determined by the thickness of the Ni_80_Fe_20_ layer, as shown in Figure 6b, with the highest output performance observed at around 10 nm thickness. This result aligns with the findings in Figure 3b.

In CV mode, the thickness of the Ni_80_Fe_20_ layer and the load resistance significantly impact sensor performance. Figure 6d illustrates the effect of load resistance in the CV mode of the PHMR sensor. The input voltage applied to the sensor is determined by the electrode size and the external resistance connected to the sensor. As a result of this load resistance effect, as shown in Figure 6e, the optimal Ni_80_Fe_20_ thickness decreases as the load resistance increases, leading to the highest output voltage. However, as seen in Figure 6f, the sensor’s sensitivity remains high even as the Ni_80_Fe_20_ layer thickness increases. Therefore, in CV mode, the design of the load resistance and the required Ni_80_Fe_20_ layer thickness may vary depending on the target sensor performance, whether prioritizing sensitivity or output voltage.

## 5. Discussion

In this study, we systematically investigated the combined effects of the driving mode and Ni_80_Fe_20_ layer thickness on PHMR sensor performance, with a focus on optimizing peak-to-peak voltage (V_p-p_). In CC mode, electron surface scattering at a thickness of 5–10 nm leads to a sharp increase in resistance, temporarily enhancing V_p-p_ before it begins to decline as bulk conduction becomes dominant. In contrast, in CV mode, sensor signals are primarily governed by the resistance difference between parallel and perpendicular magnetization states. However, experimental results indicate that the optimal V_p-p_ is observed at 25 nm, beyond which V_p-p_ declines despite the voltage being independent of sensor resistance. This suggests that voltage distribution across sensor layers plays a crucial role in performance, highlighting the need to account for resistance ratios and voltage allocation when designing PHMR sensors for CV mode applications.

Based on these findings, the optimal Ni_80_Fe_20_ thickness varies depending on the driving mode and application-specific requirements. In CC mode, a thickness of 10 nm provides the highest output signal by balancing surface scattering effects and bulk conduction, thereby maximizing the planar Hall voltage. In CV mode, 25 nm is the optimal thickness as Δ*ρ* approaches saturation. However, for thicknesses beyond this point, the reduction in sensor resistance suggests that voltage distribution across both the sensor and external load resistance significantly influences performance. While the output signal varies with thickness, sensor sensitivity strongly influenced by the exchange bias field tends to improve with increasing Ni_80_Fe_20_ thickness due to a reduction in exchange bias strength. Therefore, selecting the appropriate Ni_80_Fe_20_ thickness requires balancing signal strength and sensitivity based on specific application needs.

Therefore, selecting the appropriate Ni_80_Fe_20_ thickness requires balancing signal strength and sensitivity based on specific application needs. This study provides a systematic understanding of how Ni_80_Fe_20_ thickness and driving modes affect PHMR sensor performance, offering practical design guidelines for optimizing sensor fabrication. These insights contribute to the development of high-performance magnetoresistive sensors and spintronic devices, ensuring enhanced stability, sensitivity, and efficiency under various operational conditions.

## Figures and Tables

**Figure 1 sensors-25-01235-f001:**
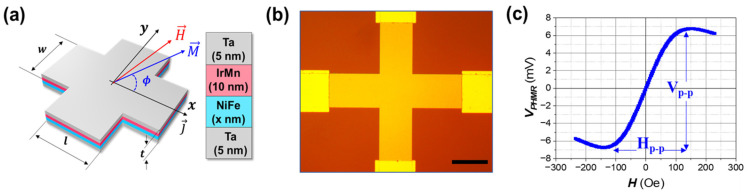
(**a**) Schematic representation of the bi-layer structure used in the PHMR sensor, consisting of Ta (5 nm)/NiFe (x nm)/IrMn (10 nm)/Ta (5 nm). The external magnetic field (H) and magnetization (M) are shown, along with the angle (*ϕ*) between them. (**b**) Optical microscope image of the fabricated cross-junction PHMR sensor with a scale bar of 200 μm. (**c**) Characterization curves of the PHMR sensor, illustrating the output voltage (V) as a function of the applied magnetic field (H).

**Figure 2 sensors-25-01235-f002:**
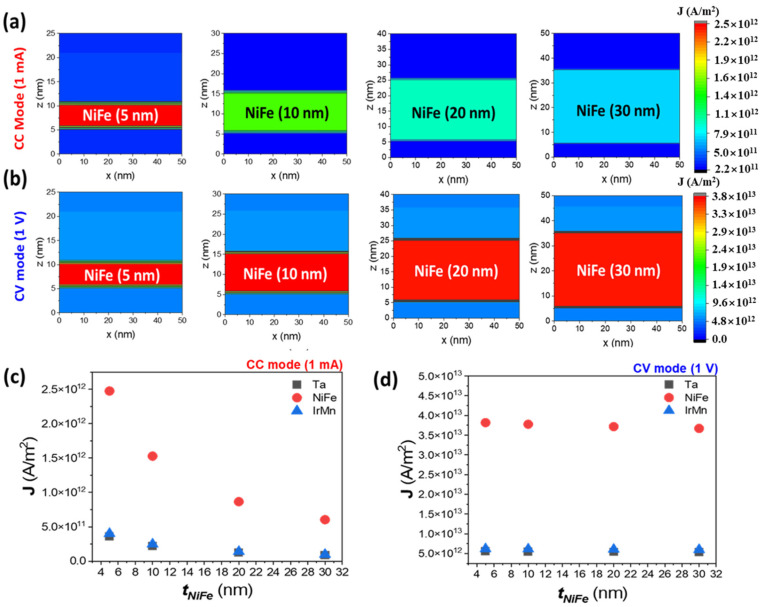
Current density distribution for Ni_80_Fe_20_ layers with varying thicknesses in CC and CV modes. (**a**) Simulated current density distributions for Ni_80_Fe_20_ thicknesses of 5 nm, 10 nm, 20 nm, and 30 nm in CC mode. The color gradient represents the magnitude of the current density, with higher densities shown in red and lower densities in blue. As the Ni_80_Fe_20_ thickness increases, the current density decreases in peak values, primarily due to shunt current effects. (**b**) Simulated current density distributions in CV mode. The current density remains relatively consistent across all thicknesses, exhibiting minimal variation as the thickness increases. (**c**) A plot of current density as a function of Ni_80_Fe_20_ thickness in CC mode and (**d**) in CV mode, with distinct markers indicating the contribution of the Ta, Ni_80_Fe_20_, and Ir_25_Mn_75_ layers.

**Figure 3 sensors-25-01235-f003:**
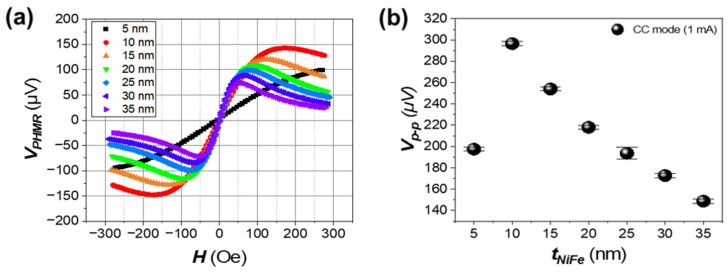
Characterization curves of the PHMR sensor signal as a function of NiFe thickness. (**a**) PHMR sensor output curves in CC Mode, visualizing the signal as the NiFe thickness varies from 5 nm to 35 nm. (**b**) Quantitative analysis of the peak-to-peak voltage difference as a function of NiFe thickness.

**Figure 4 sensors-25-01235-f004:**
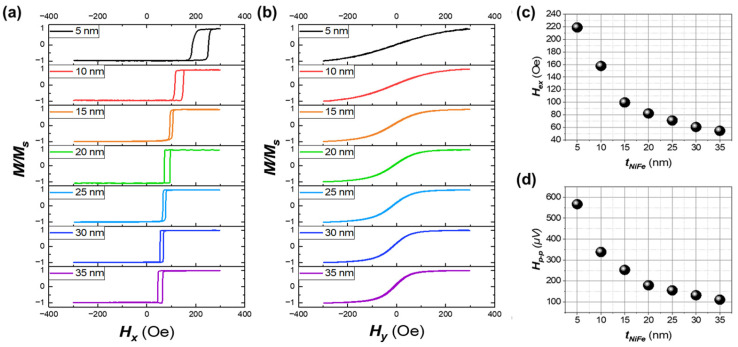
Hysteresis loops and magnetic properties of the bi-layer structure with varying NiFe thicknesses. (**a**) Easy axis and (**b**) hard axis hysteresis loops measured at room temperature using VSM. (**c**) Exchange bias as a function of NiFe thickness and (**d**) magnetic field range of the PHMR sensor with increasing NiFe thickness.

**Figure 5 sensors-25-01235-f005:**
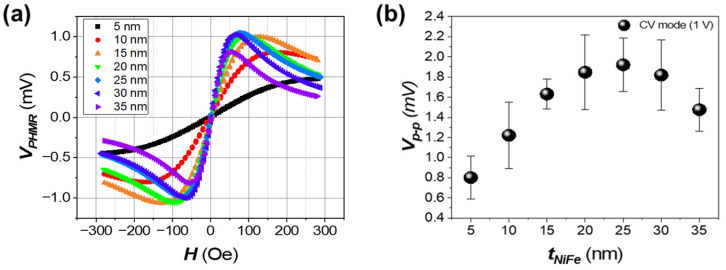
Characterization curves of the PHMR sensor signal as a function of Ni_80_Fe_20_ thickness. (**a**) PHMR sensor output curves in CV mode, visualizing the signal as the Ni_80_Fe_20_ thickness varies from 5 nm to 35 nm. (**b**) Quantitative analysis of the peak-to-peak voltage difference as a function of NiFe thickness.

**Figure 6 sensors-25-01235-f006:**
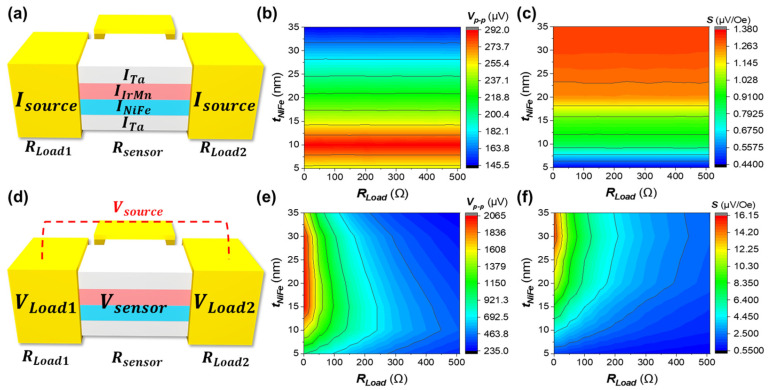
(**a**) Schematic representation of the PHMR sensor in CC mode, highlighting the shunt current effect. (**b**) Contour plot of the peak-to-peak voltage as a function of load resistance and NiFe thickness in CC mode. (**c**) The contour plot shows the sensor’s sensitivity as a function of load resistance and NiFe thickness in CC mode. (**d**) Schematic of the PHMR sensor in CV mode, illustrating the impact of load resistance. (**e**) Contour plot of the peak-to-peak voltage as a function of load resistance and NiFe thickness in CV mode. (**f**) Contour plot showing the sensitivity of the sensor as a function of load resistance and NiFe thickness in CV mode.

**Table 1 sensors-25-01235-t001:** Output characteristics of the PHMR sensor as a function of NiFe layer thickness in CC mode.

Thickness (nm)	V_p-p_ (μV)	H_p-p_ (Oe)	S (μV/Oe·mA)
5	251.2	567.1	0.443
10	291.0	338.9	0.859
15	249.2	253.4	0.983
20	223.3	180.0	1.240
25	198.2	155.6	1.274
30	173.0	132.4	1.307
35	145.9	111.2	1.312

**Table 2 sensors-25-01235-t002:** Output characteristics of the PHMR sensor as a function of NiFe layer thickness in CV mode.

Thickness (nm)	V_p-p_ (μV)	H_p-p_ (Oe)	S (μV/Oe·V)
5	801	565.0	1.64
10	1221	342.8	4.70
15	1630	260.7	7.91
20	1846	185.6	11.11
25	1920	155.3	13.06
30	1818	132.4	15.30
35	1475	111.5	14.61

## Data Availability

All data generated or analyzed during this work are included in this article.

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
