# Peer review of "Systematic Analysis of Driving Modes and NiFe Layer Thickness in Planar Hall Magnetoresistance Sensors"

_sensors, 2025, doi:10.3390/s25041235_

Round 1
Reviewer 1 Report
Comments and Suggestions for Authors
I have some questions:
1. In experiments, please give some details of experiments, for example, base pressure, the type of sputtering (DC or AC), the applied voltage levels etc.
2. Why do you choose Ni80Fe20 and Ir25Mn75?
3. Please give some references of equations.
4. In Fig.3b, the Vp-p rapidly increases with increasing t_NiFe from 5-10 nm. Why? Please Explain.
5. In Fig.4c, the Hex decreases with increasing t_NiFe. Why? Please Explain.
6. In Fig.5b, the Vp-p increases, get the highest at t_NiFe=25 nm, and decreases. Why? Please Explain.
7. What is the best or suitable case of NiFe thickness in your results? Why?
Author Response
Thank you for your insightful comments and valuable suggestions. Your questions helped us clarify key aspects of our study, including experimental details, material selection, and data interpretation. We sincerely appreciate your time and effort in reviewing our work.
Comment 1: In experiments, please give some details of experiments, for example, base pressure, the type of sputtering (DC or AC), the applied voltage levels etc.
Response 1: We have added detailed information about the experimental setup and conditions to clarify the methodology used in our study. The specific details are as follows:
1> Base Pressure and Deposition Environment:
• The base pressure during the sputtering process was maintained at 3 × 10⁻³ Torr in an argon (Ar, 99.999%) atmosphere. This low-pressure environment ensured minimal contamination during film deposition.
2> Type of Sputtering:
• A DC magnetron sputtering system was used for thin-film deposition. This method was chosen for its ability to achieve uniform film thickness and high deposition rates, particularly for metallic thin films.
3> Applied Voltage and Power:
• The applied voltage during the sputtering process ranged from 200 V to 250 V, depending on the target material. The sputtering power was set to 50 W to optimize deposition rates and uniformity.
The following revisions have been made to the manuscript between lines 92 and 98.
Comment 2: Why do you choose Ni80Fe20 and Ir25Mn75?
Response 2: We selected Ni80Fe20 (Permalloy) and Ir25Mn75 for the following reasons:
1> Ni80Fe20 (Permalloy)
• Ni₈₀Fe₂₀ is a well-known soft magnetic material widely used as the active layer in magnetoresistive sensors.
• It features high magnetic permeability, enabling rapid response to external magnetic fields, and low coercivity, minimizing magnetic hysteresis effects.
• However, even minor magnetic hysteresis can adversely affect the accuracy of sensor measurements, potentially causing inconsistencies in sensor signals under identical magnetic field conditions.
2> IrMn (Antiferromagnetic Layer)
• To address the potential hysteresis issue, Ir₂₅Mn₇₅ was employed as the antiferromagnetic layer. When paired with Ni₈₀Fe₂₀, IrMn induces exchange bias, aligning the magnetization of the Ni₈₀Fe₂₀ layer along the easy axis.
• Among various compositions of IrMn, the 25:75 ratio provides the most stable exchange bias, as confirmed in prior studies (Journal of Applied Physics 83.11 (1998): 7216-7218).
• This exchange bias minimizes hysteresis along the hard axis, ensuring linear magnetic response and improving the sensor's reliability and performance.
The reflected content has been incorporated into the manuscript from line 81 to line 87
Comment 3: Please give some references of equations.
Response 3: The equations describing the signal response of the PHMR sensor were referenced from APL Materials 10.5 (2022): 051108. This paper provides a comprehensive theoretical foundation for understanding the relationship between the magnetoresistance effect and sensor output characteristics.
The sensor characteristics under different driving modes (constant current and constant voltage) were analyzed by applying Ohm’s law to the current and voltage distribution across the sensor. This approach ensures that the derived equations align with fundamental electrical principles widely accepted in magnetoresistive sensor studies.
The reflected content has been incorporated into the manuscript in line 129
Comment 4: In Fig.3b, the Vp-p rapidly increases with increasing t_NiFe from 5-10 nm. Why? Please Explain.
Response 4: The rapid increase in Vp−p with increasing tNiFe from 5 nm to 10 nm, as observed in Fig. 3b, can be attributed to the following factor.
Regardless of the driving mode of the PHMR sensor, the output signal is significantly influenced by Δρ. As described in Journal of Applied Physics 101.5 (2007), in very thin Ni₈₀Fe₂₀ layers (e.g., 5 nm), surface scattering of electrons has a pronounced effect on resistivity, resulting in a lower Δρ (the resistivity difference between parallel and perpendicular directions). However, as the thickness increases to 10 nm, the impact of surface scattering diminishes, leading to more stable resistivity and an enhanced Δρ, which is consistent with the Fuchs–Sondheimer theory.
The reflected content has been incorporated into the manuscript from line 259 to line 267
Comment 5: In Fig.4c, the Hex decreases with increasing t_NiFe. Why? Please Explain.
Response 5: The observed decrease in Hex with increasing tNiFe, as shown in Fig. 4c, can be explained by the thickness-dependent properties of the Ni₈₀Fe₂₀/Ir₂₅Mn₇₅ bilayer system, as supported by literature (Journal of Magnetism and Magnetic Materials 305 (2006): 432-435). The following points elaborate on this phenomenon:
• Interfacial Exchange Coupling Energy: As described in the literature (Journal of Magnetism and Magnetic Materials, 192 (1999), 203–232), exchange bias (Hex) is inversely proportional to the thickness of the ferromagnetic (FM) layer. This relationship can be expressed as: Energy is the interfacial exchange coupling constant, Ms is the saturation magnetization of the FM layer, and tFM is the FM thickness. As tNiFe increases, the exchange coupling energy per unit volume decreases, resulting in a reduction in Hex.
• Reduced Interfacial Spin Coupling: For thin Ni₈₀Fe₂₀ layers, the interfacial coupling between the FM (Ni₈₀Fe₂₀) and AFM (Ir₂₅Mn₇₅) layers dominates, ensuring strong exchange bias. However, as tNiFe increases, the bulk properties of the FM layer become more significant, reducing the effective influence of interfacial spins. This leads to a weakening of the exchange anisotropy and thus a decrease in Hex.
The reflected content has been incorporated into the manuscript from line 289 to line 295
Comment 6: In Fig.5b, the Vp-p increases, get the highest at t_NiFe=25 nm, and decreases. Why? Please Explain.
Response 6: The decrease in Vp-p beyond t_NiFe=25 nm is primarily due to the saturation of Δρ and the reduction in applied voltage across the sensor layer.
• Saturation of Δρ: At tNiFe≈25nm, the resistivity difference (Δρ) between parallel and perpendicular magnetization directions approaches its saturation point. Beyond this thickness, further increases in tNiFe have minimal impact on the sensor's signal, as Δρ longer significantly changes.
• Impact on Electrical Resistance: As the thickness of the Ni₈₀Fe₂₀ layer increases, the overall electrical resistance of the sensor decreases due to reduced sheet resistance. This reduction in resistance alters the voltage distribution within the sensor and its electrodes, leading to a decrease in the voltage drop across the sensor.
• Reduction in Applied Voltage to the Sensor: The decrease in the sensor's resistance results in a lower voltage being applied across the sensor layer for a given input voltage. Consequently, the effective voltage driving the planar Hall effect is reduced, leading to a decrease in the output signal (Vp−p).
The reflected content has been incorporated into the manuscript from line 324 to line 329
Comment 7: What is the best or suitable case of NiFe thickness in your results? Why?
Response 7: The optimal thickness of the NiFe layer depends on the intended use of the sensor and its driving mode. Based on our results.
• CC Mode (Constant Current Mode):
When operating in CC mode, a NiFe thickness of 10 nm provides the optimal output signal. This is because, at this thickness, the balance between surface scattering and bulk effects is ideal for maximizing the planar Hall voltage.
• CV Mode (Constant Voltage Mode):
In CV mode, the optimal output signal is observed at a NiFe thickness of 25 nm. At this thickness, the resistivity difference (Δρ) reaches near saturation, and the voltage distribution across the sensor is most favorable for maximizing the planar Hall effect.
• Sensitivity Considerations: Unlike the output signal, the sensitivity of the sensor, defined as the ratio of the output signal to the operating range, is proportional to the size of the exchange bias field. Since the exchange bias field decreases as the NiFe thickness increases, the sensitivity generally improves with thicker NiFe layers. Therefore, selecting the appropriate thickness depends on the trade-off between signal strength and sensitivity for the specific application.
The reflected content has been incorporated into the manuscript from line 391 to line 404 in discussion
Reviewer 2 Report
Comments and Suggestions for Authors
This paper reports investigation of magnetic sensor based on planar hall effect in Py thin films. Here are several comments:
1. The constant current (CC) mode and constant voltage (CV) is very common in the measurement setup of various sensors. With CC mode thicker Py film results in lower device resistance and lower voltage on the device and thus smaller signal. Most of the contents in the abstract part of the paper is the very basic knowledge in the textbook. I suggest rewrite the whole abstract part.
2. The planar hall effect originals from the AMR effect, both of them are related to the resistance dependency on current and magnetization angle and thus the resistance anisotropy. Make more comparison of PHE and AMR.
3. Do some sensor performance characterization, e.g. equivalent magnetic noise and limit of detection. Check several references(Adv. Funct.Mater. 2023, 33, 2211752)
Comments on the Quality of English LanguageEnglish is good.
Author Response
Thank you for your valuable comments and suggestions. Your insights on the abstract, comparison between PHE and AMR, and sensor performance characterization have provided us with important directions for improving our manuscript. We appreciate your time and effort in reviewing our work and will carefully incorporate your recommendations.
Comment 1: The constant current (CC) mode and constant voltage (CV) is very common in the measurement setup of various sensors. With CC mode thicker Py film results in lower device resistance and lower voltage on the device and thus smaller signal. Most of the contents in the abstract part of the paper is the very basic knowledge in the textbook. I suggest rewrite the whole abstract part.
Response 1:
We acknowledge that the original abstract included basic textbook-level information and lacked a clear focus on the novelty and contributions of this study. Based on your suggestion, we have completely rewritten the abstract to better highlight the unique aspects of our research.
Abstract> Planar Hall magnetoresistance (PHMR) sensors have emerged as attractive candidates in sensing technology due to their simple design, cost-effectiveness, and high sensitivity. Despite their advantages, sensor performance is highly dependent on the driving mode and the thickness of the active ferromagnetic layer. Although constant current (CC) and constant voltage (CV) modes are commonly utilized in measurement setups, the interplay between these modes and the optimization of sensor performance remains insufficiently studied. This work systematically examines the effect of Ni₈₀Fe₂₀ layer thickness on the output characteristics of PHMR sensors under CC and CV modes. The findings demonstrate that in CC mode, increasing Ni₈₀Fe₂₀ thickness reduces current density, leading to diminished output signals. Conversely, in CV mode, the applied voltage mitigates the influence of thickness variations, maintaining stable signal output. This study offers a detailed analysis of the performance trade-offs associated with each driving mode, providing critical insights for the design and optimization of PHMR sensors tailored for diverse applications.
The reflected content has been incorporated into the manuscript from line 10 to line 23
Comment 2: The planar hall effect originals from the AMR effect, both of them are related to the resistance dependency on current and magnetization angle and thus the resistance anisotropy. Make more comparison of PHE and AMR.
Response 2: Thank you for your insightful comment. We agree that providing a more detailed comparison between the planar Hall effect (PHE) and anisotropic magnetoresistance (AMR) will enhance the clarity and depth of the manuscript. To address your concern, we have revised the relevant section of the manuscript to include the following points:
- Common Mechanism Between AMR and PHE:
- Both AMR and PHE are governed by the resistivity anisotropy (Δρ) and the dependence of resistance on the relative angle (ϕ) between the current direction and magnetization. This shared mechanism has been explicitly highlighted in the revised text.
- Comparison of Output Signal Characteristics:
- We have included a comparison of the output signal characteristics of AMR and PHE sensors. Specifically, AMR sensors typically exhibit higher output signals but are nonlinear at zero magnetic field due to intrinsic offset voltages. In contrast, PHE sensors demonstrate a linear response to the applied magnetic field and exhibit no offset voltages. These differences make PHE sensors particularly advantageous for applications requiring high precision and linearity.
We believe these revisions address your comment effectively and enhance the overall clarity and impact of the manuscript. Thank you for your valuable suggestion
The reflected content has been incorporated into the manuscript from line 143 to line 150
Comment 3: Do some sensor performance characterization, e.g. equivalent magnetic noise and limit of detection. Check several references(Adv. Funct.Mater. 2023, 33, 2211752)
Response 3: To address your suggestion, we have included a detailed characterization of the sensor performance, focusing on the equivalent magnetic noise and limit of detection (LOD) for both CC and CV modes. Specifically, we have added the noise spectrum analysis to Supplementary Figure S2, which illustrates the noise spectrum density of the cross-type PHMR sensor.
The analysis shows that the noise level is approximately 2.4 nV/√Hz, resulting in a magnetic resolution of 279 nT/√Hz for the 10 nm Ni₈₀Fe₂₀ thickness (CC mode) and 18.4 nT/√Hz for the 25 nm thickness (CV mode). These results demonstrate the superior magnetic resolution achievable in CV mode with optimized Ni₈₀Fe₂₀ thickness, highlighting the critical impact of driving mode and film thickness optimization on sensor performance. The corresponding details have been incorporated into the "Results" section, and we believe this addition strengthens the clarity and completeness of the manuscript.
The reflected content has been incorporated into the manuscript from line 330 to line 337
Reviewer 3 Report
Comments and Suggestions for Authors
Comment#1:
In the introduction, the study of ferromagnetism using CC and CV is not clearly explained.
Comment#2:
The current density of CC and CV is affected by the thickness of the film and needs to be described in more detail.
Comment#3:
There are many errors in punctuation and written format in the article that need to be corrected.
Comment#4:
The progress of CC and CV research on PHMR sensor needs to be elaborated, and what are the advantages of their own research compared with previous research.
Author Response
Thank you for your thoughtful comments and suggestions. Your feedback on clarifying CC and CV research, detailing current density effects, and improving the manuscript’s formatting and comparison with previous studies has been very helpful. We appreciate your time and effort in reviewing our work and will carefully incorporate your recommendations to enhance the quality of our paper.
Comment 1: In the introduction, the study of ferromagnetism using CC and CV is not clearly explained.
Response 1: Thank you for pointing out the need for a clearer explanation of the study of ferromagnetism using constant current (CC) and constant voltage (CV) modes in the introduction. We agree with your observation and have revised the introduction to better address this aspect. The revised introduction now includes the following details:
1. Single-Layer NiFe Structures:
We clarified that in single-layer NiFe structures, the sensor output signal is primarily determined by current density and magnetoresistance properties, regardless of the driving mode (CC or CV).
2. Multilayer Magnetoresistive sensor
For multilayer magnetoresistive sensors, we highlighted how the driving mode (CC or CV) impacts current density and input voltage to NiFe layer differently, leading to distinct sensor behaviors.
3. Research Gap
We explicitly stated that while CC mode has been the focus of most structural optimization studies, research on CV mode, which is crucial for industrial applications, remains limited. This study aims to address this gap by investigating the effects of driving modes on ferromagnetic material performance.
The revised introduction now provides a more comprehensive context for the study, aligning it with the research gap and emphasizing the significance of CC and CV modes in the optimization of ferromagnetic materials for PHMR sensors. We believe these changes clarify the motivation and scope of our research.
The reflected content has been incorporated into the manuscript from line 56 to line 64
Comment 2: The current density of CC and CV is affected by the thickness of the film and needs to be described in more detail.
Response 2:
We agree that the relationship between the current density and the thickness of the Ni₈₀Fe₂₀ layer in CC and CV modes requires further clarification. To address this, we have revised the manuscript, specifically in Section 4.1 of the "Results" section.
In the revisions, we have detailed the impact of driving modes on the current density distribution in the Ni₈₀Fe₂₀ layer within multilayer thin-film structures. Additionally, we have explicitly highlighted the key differences in current density behavior between CC and CV modes. These updates provide a clearer understanding of how the Ni₈₀Fe₂₀ layer thickness influences current density under different operating conditions.
(The reflected content has been incorporated into the manuscript from line 186 to line 189, and line 241~244)
Comment 3: There are many errors in punctuation and written format in the article that need to be corrected.
Response 3: We have carefully reviewed the manuscript to identify and correct errors related to punctuation and formatting.
During the review, we identified that certain subheadings (e.g., Sections 4.1 and 4.2) did not fully comply with the journal's style guidelines. These have been corrected to ensure alignment with the required format. Additionally, information in the Supplementary Materials section has been revised and formatted according to the journal's standards.
Furthermore, the Discussion section has been updated to adhere to the essential formatting requirements outlined by the journal, ensuring that it effectively communicates the key findings and their implications in a structured manner.
We appreciate your feedback and have made these changes to improve the clarity, organization, and presentation of the manuscript.
Comment 4: The progress of CC and CV research on PHMR sensor needs to be elaborated, and what are the advantages of their own research compared with previous research.
Response 4: We agree that elaborating on the progress of CC and CV research on PHMR sensors and highlighting the advantages of our research compared with previous studies is essential. To address this, we have revised the manuscript to include the following points:
4-1 Progress of CC and CV Research:
While the optimization of PHMR sensor output performance with respect to NiFe layer thickness has been extensively studied in CC mode, research involving CV mode has primarily focused on specific applications rather than thickness optimization. As such, there remains a significant gap in understanding the influence of NiFe thickness on sensor performance in CV mode.
4-2 Advantages of This Research:
In this study, we have conducted a comprehensive analysis of the output performance of PHMR sensors in both CC and CV modes. Unlike previous studies, which primarily focused on CC mode, our research addresses the influence of NiFe layer thickness on output voltage in CV mode as well. By analyzing and optimizing the thickness-dependent performance in both modes, our study provides a more complete understanding of PHMR sensor behavior, filling a critical research gap.
We have explicitly included the progress of CC and CV research in PHMR sensors in the "Introduction" section, highlighting the lack of optimization studies for CV mode. Additionally, we have emphasized the novelty and advantages of our research in the "Results and Discussion" section, particularly with respect to the comprehensive analysis and optimization of NiFe layer thickness in both modes.
The reflected content has been incorporated into the manuscript from line 56 to line 64 and line 391 to line 404.
Round 2
Reviewer 2 Report
Comments and Suggestions for Authors
I could not find obvious improvement in the revised manuscript. My original concern is mainly about the topic of this paper about the driving modes. I have to say it again. CC and CV modes are basically the same. With CV mode, the results is independent on the intrinsic resistance of the sensor, while with CC mode, the driving voltage on the device is the multiply of the driving current and the resistance. The sentences in the abstract "The findings demonstrate that in CC mode, increasing Ni₈₀Fe₂₀ thickness reduces current density, leading to diminished output signals." It is the very basic knowledge. You do not need to do any experiments to know this conclusion.
It gives me no pleasure to reject a manuscript. But I have to say that most of the information in the manuscript could be easily obtained from the textbook.
Author Response
Thank you for your insightful feedback and for highlighting the need to further clarify the key contributions of our work. We acknowledge your concern that the fundamental differences between CC and CV modes are well established. However, our study extends beyond these basic principles by providing a detailed experimental analysis of how these driving modes influence PHMR sensor performance under real-world conditions.
To address your comments more effectively, we have refined our analysis to focus on the region where Vp-p reaches its maximum, which varies depending on the driving mode. In CC mode, as expected, Vp-p increases up to a thickness of 10 nm due to enhanced resistance from surface scattering. However, beyond this point, increasing Ni₈₀Fe₂₀ thickness reduces current density, thereby decreasing the output signal. In CV mode, while the applied voltage remains independent of sensor resistance and is primarily determined by magnetization differentials, our experimental results indicate that the optimal Vp-p is achieved at 25 nm. This suggests that additional factors—such as voltage distribution effects due to reduced sensor resistance and the ratio of sensor to load resistance—play a crucial role in determining the optimal sensor thickness for maximizing performance.
To better highlight these findings, we have revised the discussion section to provide further clarification on the mechanisms underlying the thickness-dependent behavior in CV mode. Additionally, we have updated the abstract and discussion section to more clearly reflect these results and their implications. The corresponding revisions have been incorporated into the manuscript from lines 10 to 26 (Abstract) and 383 to 411 (discussion).
We believe these refinements enhance the clarity and impact of our work, and we sincerely appreciate your valuable insights, which have helped us strengthen the manuscript.
Thank you again for your time and thoughtful feedback.

Round 3
Reviewer 2 Report
Comments and Suggestions for Authors
none